# Computational Model Reveals a Stochastic Mechanism behind Germinal Center Clonal Bursts

**DOI:** 10.3390/cells9061448

**Published:** 2020-06-10

**Authors:** Aurélien Pélissier, Youcef Akrout, Katharina Jahn , Jack Kuipers , Ulf Klein , Niko Beerenwinkel, María Rodríguez Martínez 

**Affiliations:** 1IBM Research Zurich, 8803 Rüschlikon, Switzerland; 2Department of Biosystems Science and Engineering, ETH Zurich, 4058 Basel, Switzerland; katharina.jahn@bsse.ethz.ch (K.J.); jack.kuipers@bsse.ethz.ch (J.K.); niko.beerenwinkel@bsse.ethz.ch (N.B.); 3École Normale Supérieure, 75005 Paris, France; youcef.akrout@ens.fr; 4Leeds Institute of Medical Research at St. James’s, University of Leeds, Leeds LS9 7TF, UK; u.p.klein@leeds.ac.uk

**Keywords:** GC, B cell receptor, somatic hypermutation, nucleotide sequence, antibody affinity, affinity maturation, immunoglobulin, clonal competition, clonal burst, clonal dominance, stochastic models, affinity models

## Abstract

Germinal centers (GCs) are specialized compartments within the secondary lymphoid organs where B cells proliferate, differentiate, and mutate their antibody genes in response to the presence of foreign antigens. Through the GC lifespan, interclonal competition between B cells leads to increased affinity of the B cell receptors for antigens accompanied by a loss of clonal diversity, although the mechanisms underlying clonal dynamics are not completely understood. We present here a multi-scale quantitative model of the GC reaction that integrates an intracellular component, accounting for the genetic events that shape B cell differentiation, and an extracellular stochastic component, which accounts for the random cellular interactions within the GC. In addition, B cell receptors are represented as sequences of nucleotides that mature and diversify through somatic hypermutations. We exploit extensive experimental characterizations of the GC dynamics to parameterize our model, and visualize affinity maturation by means of evolutionary phylogenetic trees. Our explicit modeling of B cell maturation enables us to characterise the evolutionary processes and competition at the heart of the GC dynamics, and explains the emergence of clonal dominance as a result of initially small stochastic advantages in the affinity to antigen. Interestingly, a subset of the GC undergoes massive expansion of higher-affinity B cell variants (clonal bursts), leading to a loss of clonal diversity at a significantly faster rate than in GCs that do not exhibit clonal dominance. Our work contributes towards an in silico vaccine design, and has implications for the better understanding of the mechanisms underlying autoimmune disease and GC-derived lymphomas.

## 1. Introduction

Germinal centers (GCs) [1,2,3] are specialized microanatomical structures that emerge within the secondary lymphoid organs upon infection or immunization. A GC typically contains up to a few thousand B cells that rapidly proliferate and mutate the immunoglobulin genes of their B cell receptors (BCRs). GCs play a central role in mounting an effective immune response against infectious pathogens, and failures within their tightly regulated environment can lead to the development of autoimmune diseases [4] and cancer [5]. Hence, a detailed investigation of the processes that regulate the dynamics of the GC reaction is of crucial importance to understand both healthy and pathological immune responses.

GCs are histologically divided into two different compartments, the dark zone (DZ) and light zone (LZ), which play distinct and important roles in the maturation and selection of B cells [6]. The DZ consists primarily of a tight cluster of highly proliferative B cells, known as centroblasts (CBs), while the LZ is less compact and more diverse, including GC B cells, known as centrocytes (CCs), antigen-presenting follicular dendritic cells (FDCs) and a smaller population of T follicular helper cells (T${}_{FH}$). A GC reaction is initiated when naive B cells are activated by T${}_{FHs}$. Upon activation, naive B cells initiate a phase of monoclonal expansion characterized by very rapid proliferation rates and somatic hypermutation (SHM) of their immunoglobulin genes [7,8]. After a few days, GCs become polarized into the DZ and LZ, and CBs migrate to the LZ, where they undergo a phase of selection. CCs in the light zone are committed to apoptosis and compete for survival signals delivered by T${}_{FHs}$. To receive such signals, CCs must first acquire antigen from FDCs. Higher affinity CCs can capture larger amounts of antigen, which is then engulfed, digested into peptides and returned to the B-cell surface bound to the major histocompatibility MHC class II. T${}_{FHs}$ that bind with enough strength to the peptide–MHC complex (pMHC) deliver signals that stop apoptosis, upon which a CC can leave the GC and terminally differentiate into a plasma cell (PC), responsible for secreting antibodies, or into a long-lived memory B cell (MBC) that keeps memory of past infections and can rapidly respond to repeated antigen exposure. Low affinity cells that do not receive enough T${}_{FH}$ signaling are eliminated by apoptosis in a process that replicates Darwinian evolution at the cellular level.

In addition, a fraction of CCs return to the DZ for additional rounds of cell division and BCR maturation [9]. The speed of the cell cycle in the DZ is regulated by the amount of signalling received from the T${}_{FHs}$ [10], likely by upregulating cell-cycle regulators such as MYC [11,12,13]. As a result, high-affinity cells that receive strong T${}_{FHs}$ signals undergo accelerated cell cycles and can replicate up to 6 times, while lower affinity cells that capture less antigen divide fewer times [14]. The regulation of the cell cycle critically contributes to the selection and clonal expansion of high-affinity cells as well as to the observed progressive decline of clonal diversity in at least a subset of GCs [15], although detailed quantitative models are still needed to understand mechanisms behind clonal evolution, competition and clonal burst induction.

**Quantitative modelling of GCs**: At the molecular level, the intracellular mechanisms associated with regulation of the B cells, T${}_{FHs}$ and FDCs interactions implicates more than 100 transcription factors [16], most of which interact in highly regulated non-linear networks [17], making the precise quantitative modeling of GC reaction tremendously complex. As GCs are stochastic systems that display a high level of variability even within the same lymph node of the same individual [18], mathematical models have been widely used to deepen our understanding of the cellular and molecular processes characterising these complex dynamic systems [19]. In particular, multi-scale stochastic [20] and spatial agent-based models have been proposed [21,22,23]. The advantage of such models is their faithful replication of the probabilistic interactions between the different cellular populations in the GC. Spatial models can capture the spatial dynamics and cellular flow between the two GC compartments, although they are encumbered with several methodological challenges and computational complexity. In comparison to spatial models, stochastic models offer fast and efficient computation of the main statistical properties of the GC with the theoretical guaranties of convergence to the exact probabilistic cellular distributions. Alternatively, computational models based on ordinary differential equations (ODEs) tracking the evolution of individual cells have also been proposed, and concluded that there is limited correlation between subclone abundance and affinity [24]. Other ODE models [25] were used to look at clonal diversity with a simple birth, death and mutation model.

While these models have successfully reproduced the GC dynamics and B cell maturation process, the accurate investigation of clonal diversity and burst emergence requires detailed modelling of the affinity maturation process based on more realistic representations of the BCRs, where the impact of newly acquired somatic mutations on the antigen binding capacity can be assessed. Advances in next-generation sequencing of immunoglobulin genes (Ig-seq) have revealed the dynamics of BCR sequence diversification across different B cell types in healthy and antigen-stimulated B cell donors [26]; however, structural information about the BCR, which is crucial to accurately model antibody binding capability [27], is not available in a sequencing experiment. Preliminary work on BCR structural representations that model the BCR as a short amino acid chain of (∼10 AAs) on a rectangular lattice has recently been developed [28]. Such detailed modelling comes however at a high computational cost—a single GC simulation takes a few hours, which currently limits its application to GC simulations.

In this paper we build on a previous multi-scale GC model, where the cellular interactions of B cells, T${}_{FH}$ and FDCs were modelled stochastically using the Gillespie algorithm [20]. We improve the model by adding an abstract molecular representation of the BCR based on its nucleotide sequence that allows us to track the effect of SHM on the BCR affinity through time. Our quantitative GC model is driven by 10 stochastic cellular interactions, which are parameterized using recent experimental characterizations of the GC dynamics. We expand the previous model by adding new regulatory mechanisms, such as centroblast division regulation by MYC, or centroblast apoptosis due to nonproductive BCRs, and use the extended model to explore different differentiation scenarios of B cells. Our sequence-based BCR representation enables a detailed investigation of clonal evolution using phylogenetic trees [29], and an analysis of the factors that determine the emergence of clonal dominance. Our paper summarizes the current quantitative knowledge about the GC responses, and gives additional information about clonal diversity and B-cell affinity maturation, which is particularly challenging to obtain from experimental data.

## 2. Methods: A Probabilistic Model of GC Dynamics

### 2.1. Stochastic Model of the GC

Our GC model includes reactions that occur in the low frequency regime where stochastic effects cannot be neglected, and hence cannot be simulated using ordinary differential equations. Instead, we use the formalism of chemical kinetics, where each reaction represents a stochastic process between the following reactants: naive B cells (NB), centroblasts (CB), centrocytes (CC), selected centrocytes (CC${}_{\mathrm{sel}}$), memory B cells (MBC), plasma cells (PC), T follicular helper cells (T${}_{FH}$), and follicular dendritic cells (FDC). The set of reactions are given in Table 1, and graphically depicted in Figure 1.

To model the set of equations, we use the Gillespie algorithm [30], an algorithm that generates statistically correct trajectories of a stochastic system of equations according to simple mass-action kinetics. Briefly, at each time step, a propensity $p_{i}$, or instantaneous probability that a reaction takes place, is computed as $p_{i}=\prod_{j=1}^{s}N_{i_{j}}*r_{i}$, where $N_{i_{j}}$ are the number of reactants involved in reaction *i* and $r_{i}$ is the reaction rate of the channel. The algorithm iteratively draws two random numbers: The first one is used to define a time interval according to an exponential distribution, and the second one is used to choose the reaction that takes place in that time interval. It can be proven that the Gillespie algorithm converges to the exact probabilistic cellular distribution of all reactants. Furthermore, it is computationally cheap compared to alternative simulation methods, and has been extensively applied to model biomolecular interactions [31,32,33].

The standard Gillespie algorithm assumes that all particles are identical, and hence, it cannot be applied to model cellular systems where individual cellular properties need to be distinguished, such as the amount of antigen acquired by a CC, or its BCR sequence which determines its affinity and probability of capturing antigen. To overcome this limitation, a modified Gillespie [20] algorithm was developed that enables the efficient simulation of particles with individual properties by exploiting Poisson thinning, a process whereby propensities that include undesired reactions are computed and subsequently rejected. The modified algorithm was applied to simulate the GC reaction where it demonstrated speed and accuracy. Here we use the modified Gillespie algorithm to account for the following reactions: (i) CB division, (ii) CB migration, (iii) CC-T${}_{FH}$ binding, (iv) CC competition for T${}_{FH}$ help, and (v) CC spontaneous unbinding. A brief description of each one of the cellular interactions follows:

**GC initiation and founder cells**: $\mathrm{NB}\overset{r_{\mathrm{activation}}}{\to }\mathrm{CB}$

A GC starts roughly 4 days after initial exposure to an antigen, after naive B cells are activated by exposure to exogenous antigen within the lymph nodes. Early interclonal competition for T cell signals selects a subset of B cells with the highest relative affinity to enter the GC reaction, contributing to the early establishment of the oligoclonality in the GC [34]. Our GC model starts empty at day 4 after immunization, after which time B cells start to enter the GC reaction at a rate of $r_{\mathrm{activation}}$ per hour during the next 5 days (implying $5\;\cdot \;24\;\cdot \;r_{\mathrm{activation}}$ seeder cells) [22,35]. The propensity of this reaction can be computed as:$$\mathrm{propensity}=\left\{\begin{array}{cc} r_{\mathrm{activation}}\;,\; & \mathrm{between}\;\mathrm{day}\;4\;\mathrm{and}\;\mathrm{day}\;9,\\ 0\;,\;\; & \mathrm{otherwise}. \end{array}\right.$$

In our simulation, we use a division counter [14] that takes values between 0 and 6, and is reduced by one each time a B cell divides. Once the division counter falls to zero, the cell can no longer divide and migrates to the LZ. Founding cells are initialized with the maximum division counter, i.e., 6. We assume an initial affinity for BCRs ≃0.4 (see Appendix A for discussion about the choice of parameters).

**Centroblast division**: $\mathrm{CB}\overset{r_{\mathrm{division}}}{\to }2\mathrm{CB}$

CBs proliferate up to 6 times in the DZ [14], and randomly acquire mutations in their BCR genes through SHM after each division. Similarly to seeder cells, we endow CBs with a division counter, which allows them to divide up to 6 times depending on the amount of T${}_{FH}$ help received. Cells are only allowed to migrate to the LZ once the counter has fallen to 0. The propensity of this reaction is:$$\mathrm{propensity}=\left\{\begin{array}{cc} N_{\mathrm{CB}}\;\cdot \;r_{\mathrm{division}}\;,\; & \mathrm{if}\;\mathrm{division}\;\mathrm{counter}\;\mathrm{is}\;>\;0,\\ 0\;, & \mathrm{if}\;\mathrm{division}\;\mathrm{counter}\;\mathrm{is}\;0. \end{array}\right.$$

Each daughter CB undergoes apoptosis ($\mathrm{CB}\overset{}{\to }\emptyset$) with probability δ per mutation (see Section 2.2).

**Centroblast migration DZ → LZ**: $\mathrm{CB}\overset{r_{\mathrm{migration}}}{\to }\mathrm{CC}$

Once a CB division counter drops to 0, the cell can migrate to the LZ, where its regulatory program significantly changes. The propensity of this reaction is:$$\mathrm{propensity}=\left\{\begin{array}{cc} N_{\mathrm{CB}}\;\cdot \;r_{\mathrm{migration}}\;,\; & \mathrm{if}\;\mathrm{division}\;\mathrm{counter}\;\mathrm{is}\;0,\\ 0\;, & \mathrm{if}\;\mathrm{division}\;\mathrm{counter}\;\mathrm{is}>\;0. \end{array}\right.$$

Experiments suggest that class II MHC molecules are rapidly turned over on DZ B cells by ubiquitin-mediated degradation, ensuring that the peptide-MHC II density is reset after every LZ-DZ cycle [36]. Therefore, we assume that the number of antigen-MHC complexes is reset to 0 at the time of migration, which ensures fair selection against other newly activated CCs in the LZ.

**Centrocyte antigen uptake from FDC**: CC + FDC $\overset{r_{\mathrm{FDC}\text{:}\mathrm{CC}}}{\to }$ CC(pMHC) + FDC

Centrocytes diffuse in the LZ until they encounter an FDC. Since the encounter rate is inversely proportional to the LZ volume, $r_{\mathrm{FDC}-\mathrm{CC}}\propto 1/{\mathrm{vol}}_{\mathrm{LZ}}$, i.e., it is more likely for a CC to encounter a given FDC when the GC volume is small. As ${\mathrm{vol}}_{\mathrm{LZ}}$ is proportional to the number of CCs, $N_{\mathrm{CC}}$, the propensity can be written as:$$\mathrm{propensity}=r_{\mathrm{FDC}\text{:}\mathrm{CC}}\;\cdot \;N_{\mathrm{FDC}}\;\cdot \;N_{\mathrm{CC}}\propto N_{\mathrm{FDC}}\;.$$

We assume that antigen uptake occurs when a CC encounters an FDC, upon which, the CC acquires antigens in an affinity-dependent manner and re-exposes on its surface as a pMHC complex. Mature B cells have been reported to carry up to $10^{5}$ BCRs on their cell surface [37], and FDCs present a very large amount of antigen [38], hence we assume that there is no competition for antigen and a B cell can acquire as much antigen as the affinity of its BCR allows. Therefore, upon contact with an FDC, the pMHC complex of the centrocyte is updated with pMHC = affinity${}_{\mathrm{CC}}$. In the event of the CC having previously acquired antigens from another FDC, the pMHC complex is replaced. The interaction of GC B cells with FDCs is of the order of a second [6,39], and hence, we do not explicitly model the binding time between B cells and FDCs. Alternative models that take into account spatial motion have included a refractory time during which a B cell cannot attempt antigen binding again [22]. This interval is necessary to give time for the CC to diffuse away from the FDC and prevent repetitive binding. However this is not necessary in our model, as the Gillespie algorithm models cellular distributions, and the probability of the same CC–FDC pair binding more than once is very small, ∼ $\left(r_{\mathrm{FDC}\text{:}\mathrm{CC}}/\sum_{i}\mathrm{propensity}\;{(\mathrm{reaction}}_{i}\right){)}^{2}$.

**Centrocyte–T${}_{FH}$ binding**: $\mathrm{CC}+T{}_{FH}\overset{r_{\mathrm{TC}\text{:}\mathrm{CC}}}{\to }\left[\mathrm{CCTC}\right]$

After acquiring antigen, CCs diffuse into the LZ until they encounter a free T${}_{FH}$. Upon encounter, CCs attempt to bind their pMHCs ligands to the T cell receptors expressed on the surface of the T${}_{FHs}$. The propensity is derived in the same manner as for the encouters with FDCs:$$\mathrm{propensity}=\left\{\begin{array}{cc} r_{\mathrm{TC}\text{:}\mathrm{CC}}\;\cdot \;{\left(N_{T_{\mathrm{FH}}}\right)}_{\mathrm{unbound}}\;\cdot \;N_{\mathrm{CC}}\;,\; & \mathrm{if}\;\mathrm{pMHC}\;>\;0,\\ 0\;, & \mathrm{if}\;\mathrm{pMHC}\;=\;0. \end{array}\right.$$

Regarding the number of unbound T${}_{FHs}$, T${}_{FHs}$ enter and leave the GC during the whole reaction [40]. Therefore, our model assumes that a homeostatic flow of T${}_{FHs}$ has been established that leads to an approximately constant density of T${}_{FHs}$ during the whole reaction [18] (see Section 2.4 for a more detailed discussion). Since the volume of the LZ is proportional to the number of CCs, we can write:$${\left(N_{T_{\mathrm{FH}}}\right)}_{\mathrm{unbound}}+N_{\left[\mathrm{CCTC}\right]}={\left(N_{T_{\mathrm{FH}}}\right)}_{\mathrm{total}}=\alpha_{\mathrm{TC}}\;\cdot \;N_{\mathrm{CC}}$$

To implement this constraint, we dynamically adjust the number of T${}_{FHs}$ at every iteration of the Gillespie algorithm such that the above equation is always verified.

**Centrocyte competition for T${}_{FH}$ help**: $\left[{\mathrm{CC}}_{1}\mathrm{TC}\right]+{\mathrm{CC}}_{2}\overset{r_{\mathrm{TC}\text{:}\mathrm{CC}}}{\to }{\mathrm{CC}}_{1}+\left[{\mathrm{CC}}_{2}\mathrm{TC}\right]$

Two-photon microscopy imaging reveals that the majority of T${}_{FHs}$ in GCs are not engaged in stable interactions with B cells, but instead, most contacts between GC B cells and T${}_{FHs}$ are of short duration (<5 min) [6]. In our model, we replicate this phenomenon by introducing a competition between the CCs to bind to T${}_{FHs}$. We assume that, if during the time a CC${}_{1}$ is bound to a T${}_{FHs}$ a second CC${}_{2}$ attempts to bind to the same T${}_{FH}$, the CC with the larger amount of pMHC will remain bound and the other CC will be rejected [20]. The propensity can be written as: $$\mathrm{propensity}=\left\{\begin{array}{cc} r_{\mathrm{TC}\text{:}\mathrm{CC}}\;\cdot \;N_{\left[\mathrm{CCTC}\right]}\;\cdot \;N_{\mathrm{CC}}\;,\; & \mathrm{if}\;\mathrm{pMHC}\left({\mathrm{CC}}_{2}\right)>\mathrm{pMHC}\left({\mathrm{CC}}_{1}\right),\\ 0\;, & \mathrm{otherwise}. \end{array}\right.$$

As a result of this competition, a CC can interact with a T${}_{FH}$ multiple times for shorter amounts of time in our model. We assume that all these signals are cumulative and contribute towards fate decision [23,41].

Furthermore, a CC substitution reaction happens in the timescale of a diffusion process, which is typically much faster than the timescale of the GC reaction. Hence, we neglect the time associated with the unbinding of CC${}_{1}$ and binding of CC${}_{2}$, and assume that this reaction has the same rate as a standard CC–T${}_{FH}$ binding.

**Centrocyte–T${}_{FH}$ spontaneous unbinding**: $\left[\mathrm{CCTC}\right]\overset{r_{\mathrm{unbinding}}}{\to }{\mathrm{CC}}_{\mathrm{sel}}+T{}_{FH}$

A CC bound to a T${}_{FH}$ can unbind spontaneously after having received a sufficient amount of T${}_{FH}$ help: $$\mathrm{propensity}=\left\{\begin{array}{cc} N_{\left[\mathrm{CCTC}\right]}\;\cdot \;r_{\mathrm{unbinding}}\;,\; & \mathrm{if}\;\mathrm{it}\;\mathrm{was}\;\mathrm{in}\;\mathrm{contact}\;\mathrm{with}\;T_{FH}\;\mathrm{for}\;\mathrm{more}\;\mathrm{than}\;30\min,\\ 0\;, & \mathrm{otherwise}. \end{array}\right.$$

Once a CC and T${}_{FH}$ unbind after the CC has received enough T${}_{FH}$ signal, the T${}_{FH}$ cell is released and the CC switches to the state selected.

**Centrocyte apoptosis**: $\mathrm{CC}\overset{r_{\mathrm{apoptosis}}}{\to }\emptyset$

All CCs are subjected to a timed program regardless of their BCR affinity, and undergo apoptosis with rate $r_{\mathrm{apoptosis}}$ if not rescued by T${}_{FH}$ signals:$$\mathrm{propensity}=r_{\mathrm{apoptosis}}\;\cdot \;N_{\mathrm{CC}}\;.$$

**Centrocyte recirculation**: ${\mathrm{CC}}_{\mathrm{sel}}\overset{r_{\mathrm{recirculate}}}{\to }\mathrm{CB}$

Selected CCs can recircule into the DZ for additional rounds of division and affinity maturation. Their division counter is reset to 1–6 depending on the amount of T${}_{FH}$ signals received (Section 2.6).
$$\mathrm{propensity}=N_{{\mathrm{CC}}_{\mathrm{sel}}}\;\cdot \;r_{\mathrm{recirculate}}\;.$$

**Centrocyte exit**: ${\mathrm{CC}}_{\mathrm{sel}}\overset{r_{\mathrm{exit}}}{\to }\mathrm{MBC}\;\mathrm{or}\;\mathrm{PC}$

Similarly, CC${}_{\mathrm{sel}}$ can terminally differentiate as a PC or MBC with the following propensity:$$\mathrm{propensity}=N_{{\mathrm{CC}}_{\mathrm{sel}}}\;\cdot \;r_{\mathrm{exit}}\;.$$

We assume that once a CC has been selected to differentiate, it becomes a PM or a MBC in a pMHC-dependent manner. In Section 2.7, we computationally investigate three alternative scenarios to determine fate choice.

### 2.2. Cell Fate after Somatic Hypermutation

Most of the variability of an antibody is concentrated into three hypervariable loops called complementarity determining regions, CDR1, CDR2, and CDR3, which are responsible for the extensive range of antigens with which antibodies can interact [42]. A mutation in the CDR typically modifies the antigen binding properties of the BCR [43], while a mutation in the surrounding framework regions (FWR) might compromise its structural support and potentially lead to apoptosis. Not surprisingly, CDR chains show higher variability than FWR chains [44], although this might be due to posterior selection [6].

We model the effect of a mutation using a decision tree [24]. In this representation, a nucleotide change can either: (1) be neutral or silent, i.e., the mutation has no impact on affinity, with probability σ; (2) result in a nonproductive BCR that leads to apoptosis with probability δ; or (3) change the affinity of the BCR with probability β=1−σ−δ (Figure 2). In our simulation, we assume that a mutation can happen in both the FWR and CDR regions, although only mutations in the CDR region can lead to a change in affinity. To model the effect of a mutation, we draw two random numbers. The first number determines the number of single nucleotide changes in the whole BCR sequence according to a binomial distribution with parameters $N_{\mathrm{BCR}}=660$ [45] and $p_{\mathrm{SHM}}=1\times 10^{-3}$ [46]. The second random number determines the impact of each mutation according to Figure 2.

### 2.3. B Cell Receptor Affinity to Epitote

To model changes in affinity, we only consider the sequence $S_{\mathrm{CDR}}$ of the CDR region of each BCR. Each sequence $S_{\mathrm{CDR}}$ is composed of $N_{\mathrm{CDR}}$ sites that can take 4 possibles values ∈[A,T,G,C]. To simplify sequence comparison, we further assume that all the BCRs sequences are of the same length ($N_{\mathrm{CDR}}=25$). We define the affinity of a sequence by using the normalized Hamming distance between the sequence $S_{\mathrm{CDR}}$ and the optimal CDR sequence $S_{0}$ to bind the target antigen, as follows:(1)$$a_{\mathrm{CDR}}=1-\frac{\left|\right|S_{\mathrm{CDR}}-S_{0}{\left|\right|}_{H}}{N_{\mathrm{CDR}}}\;.$$

While this representation does not consider the structural properties of the BCR, it has the advantage of modelling the effect of individual nucleotide changes and being computationally very cheap to compute, which enables the running of exhaustive simulations of the GC with a realistic number of cells. Furthermore, Equation (Equation 1) is bounded by $0\leq a_{\mathrm{CDR}}\leq 1$, and hence, $a_{\mathrm{CDR}}$ is endowed with the general properties expected of any physiologically relevant affinity function. For instance, it becomes probabilistically less likely to increase the affinity of a sequence that is close to the upper bound. Conversely, if a sequence exhibits low affinity, there is a wide range of mutations that can significantly increase the affinity. Our definition of affinity has strong similarities with previously postulated affinity measures [47], with the important difference being that we use the Hamming distance instead of the Euclidean distance, which is more adequate to work with strings of characters (nucleotides). It also agrees with other postulated properties of the affinity function such as: homogeneity, i.e., the function applies equally well to all regions of the shape space; complementarity, i.e., the optimal sequence can also be represented as a sequence, where shortest distance to the antigen sequence corresponds to greater affinity; and smoothness, i.e., small changes in the distance correspond to small changes in the affinity [48].

### 2.4. T${}_{FH}$ Dynamics

Two-photon microscopy experiments have shown that T${}_{FHs}$ exist in a state of dynamic exchange between different GCs in a lymph node, and newly activated T${}_{FHs}$ can invade pre-established GCs and contribute to B cell selection and plasmablast differentiation [40]. Furthermore, T${}_{FHs}$ also proliferate [18], which results in cell count variability across time points. However, the ratio of GC B cells to T${}_{FHs}$ in the LZ appears to be maintained at a relatively constant ratio of ∼7:1 through the entire GC reaction [18]. We also note that a constant ratio of CC:T${}_{FHs}$ results in an approximately constant probability of a CC encountering a T${}_{FH}$ through the whole GC response.

In a typical GC, T${}_{FHs}$ are activated by a variety of different antigens, and hence, CCs can only successfully bind to a subset of the available T${}_{FHs}$. Since our model assumes that only one type of antigen is presented by the FDCs and acquired by CCs, we decrease the experimentally determined CC:T${}_{FHs}$ ratio to a lower number, which we optimise using time resolved measurements on GC kinetics data (Section 2.10). Our estimation is that the optimal effective ratio is ≈1/46, suggesting that only 15% of measured T${}_{FHs}$ are specialized to a specific antigen.

### 2.5. T${}_{FH}$ Survival Signals

The signaling interaction between a CC and a T${}_{FH}$ is non-linear: ICOSL on B cells promotes upregulation of CD40L on T cells, which in turn further upregulates ICOSL on the B cell, creating a positive feedback loop that promotes a mode of intimate contact in which a large area of the B and T cell surfaces are juxtaposed [49]. Thus, small differences in pMHC density between B cells of different affinities can be nonlinearly amplified into large differences in signaling that allow for a more efficient selection of high affinity B cells. To reproduce this behavior, we describe the strength of the binding signal using a family of monotonically increasing functions with parameter *n*, such as:$$\mathrm{signal}\;\mathrm{strength}\;\left(\mathrm{pMHC}\right)=\exp({\mathrm{pMHC}}^{n})-1\;,$$
where the parameter *n* controls the strength of the nonlinear interaction: higher values of *n* make it harder for cells with low or moderate affinity to acquire T${}_{FH}$ signals (see the Appendix A for a more detailed discussion). At every unbinding event, the cumulative signal of the CC acquired from T${}_{FH}$ is incremented with:$$\Delta T_{\mathrm{help}}=\mathrm{time}\;\mathrm{in}\;\mathrm{contact}\;\mathrm{with}\;T_{FH}\;\cdot \;\mathrm{signal}\;\mathrm{strength}\left(\mathrm{pMHC}\right)\;.$$

Thus, in our model, the total amount of help received from a T${}_{FHs}$ is the cumulative signal obtained by aggregating the signals received through each encounter, however small, that a CC has with a T${}_{FH}$ in the LZ. In practice, most encounters are of short duration and typically increase the cumulative T${}_{FHs}$ help by only a small amount (70% of the bindings last for less than 5 min). This mimics the observation that a CC probes and attempts to bind to many different T${}_{FHs}$ endowed with different receptors, before receiving a substantial amount of signalling from a T${}_{FH}$ with a compatible receptor.

### 2.6. Regulation of the Cell Cycle

The process of acquiring T cell help ends when a CC unbinds spontaneously from a T${}_{FH}$ after having received signals for ∼30 min, upon which, the CC is considered selected and can either leave the GC and terminally differentiate into a PC or a MBC, or recirculate into the DZ for further rounds of division and BCR maturation. Cells that recirculate into the DZ regulate their cell cycle speed according to the amount of T${}_{FHs}$ help received, with cells that have received strong signals being able to replicate up to 6 times [10,14]. The regulation of the cell cycle is likely to be induced by MYC [11,13], a cell-cycle regulator that is upregulated by T${}_{FH}$ help in cells selected for reentry into the DZ [12]. MYC is also known to play a critical role in the regulation of division bursts in the DZ [11].

To reproduce the regulation of the cell cycle, we use a division counter [14] that takes values between 1 and 6 depending on the amount of T${}_{FH}$ help received. To implement the division counter, we first run a short simulation to precompute the distribution of cumulative help from T${}_{FH}$. This initial distribution is kept fixed throughout the GC simulation and used to define 6 equal percentiles, such that a cell in the nth percentile undergoes *n* divisions (Figure 3). Such definition corresponds to an average of ∼3.5 divisions per recirculating cell in our model, very close to the experimentally determined average of 3 divisions per recirculating cell [14]. Besides increasing in number due to the higher number of cellular divisions, high-affinity cells also acquire a higher number of mutations than those that undergo fewer divisions [14], which enhances their chances of developing more effective BCRs.

### 2.7. Plasma Cell and Memory B Cell Differentiation

Upon exit from the GC, B cells terminally differentiate into PCs or MBCs. Temporal data has demonstrated the existence of a temporal switch, where MBCs tend to be produced during the early phases and PCs are produced at later times of the response [50]. The precise mechanism that determines whether a cell differentiates into a PC or MBC is not completely understood [1], although recent work suggests that affinity to antigen plays a central role in selecting the cells that enter the PC pool, while lower affinity selected B cells progress to MBCs [51]. In this scenario, T${}_{FH}$ help stimulates CD40 signaling and IRF4 transcription, which in turn regulates, in a dose-dependent manner, the choice between differentiating into a PC or cycling back into the DZ [52]. This model has been computationally investigated using a multi-scale probabilistic model and shown to lead to good agreement with the experimentally determined GC output [20]. Regarding the differentiation into MBCs, the determinants that predispose cells to enter the MBC pool or the markers that identify cells committed to the MBC lineage are not known. A fraction of LZ B cells with lower affinity are prone to progress to resting MBCs in a Bach2 TF-dependent manner [53,54], although high-affinity cells might also become MBCs. This suggests that stochastic effects might be at play in the selection of the MBC fate.

We use our model to investigate alternative differentiation scenarios. Similarly to previous work, we assume that IRF4 regulates cell fate in a dose-dependent manner, where high-affinity cells that express high amounts of IRF4 are directed to the PC pool, and medium affinity cells either cycle back into the DZ or differentiate as a MBC. Once a cell is positively selected, we simulate and analyse three possible differentiation scenarios:**Antigen-driven differentiation model (model 1)**: In this first scenario, we consider that affinity determines B cell fate in a deterministic manner. Namely, there is a threshold pMHC${}_{\mathrm{thresh}}$ above which a CC becomes a PC. Below that threshold, a cell becomes a MBC.**Dynamic antigen-driven differentiation model (model 2)**: A more sophisticated hypothesis is that T cells progressively become less sensitive to the pMHC complex and, as the average affinity of B cells increases in the GC, a larger amount of pMHC is needed to induce the same amount of IRF4. In the second scenario, the choice between PC and MBC is also deterministic, although according to a dynamic threshold that increases linearly with time, i.e., pMHC${}_{\mathrm{thresh}}$ = affinity${}_{GC}\left(t-t_{0}\right)$ + a${}_{0}$.**Stochastic differentiation model (model 3)**: Finally, we also investigate a probabilistic scenario where, once a B cell has been selected for differentiation, it becomes a PC or MBC according to some fixed probabilities that depend on the amount of expressed pMHC as follows: $p_{\mathrm{MBC}}=e^{-k(\mathrm{pMHC}-p_{0})}$ and $p_{\mathrm{PC}}=1-p_{\mathrm{MBC}}$.

In all three scenarios, apoptosis occurs irrespective of affinity. Namely, apoptosis is modelled in the Gillespie algorithm as a reaction channel with constant kinetic rate, and hence, its probability of happening only depends on the total number of CCs. We note, however, that CCs carrying high-affinity BCRs are more likely to be positively selected and differentiate or recirculate, while lower affinity CCs are more likely to stay in the GC and undergo apoptosis, which might result in an indirect association between low affinity and apoptosis [55]. Regarding recirculation, it happens stochastically according to a Gillespie channel of constant kinetic rate once a CC has been selected, and hence, recirculation after selection is an affinity-independent event. In all scenarios, the amount of received T cell signaling is only important to stop apoptosis and to decide on the number of divisions for cells that recirculate into the DZ. In that respect, all models are based on antigen-driven selection mechanisms.

### 2.8. Phylogenetic Trees Representation

We use phylogenetic trees to visualize the evolution of the CDR sequences through SHM. In such a representation, each founder cell, associated with a unique V, D and J combination, defines the unmutated germline of a new tree, and newly acquired mutations are represented as downstream nodes. Different approaches to represent clonal trees are possible. Here, we chose to associate each leaf with a sequenced cell and each internal node with an unique mutation of the BCR. Using our simulated data, it is trivial to reconstruct lineage trees, as we can follow each founder cell and its progeny through the whole GC reaction. Then, the topological properties of each phylogenetic tree, such as number of nodes, average depth of the leaves, trunk length, etc provide important information about the affinity maturation [29,56], and can be used to compare simulated trees to trees inferred from experimental data.

To construct trees from experimental data, i.e., repertoire sequencing data, we first align the variable (V), diversity (D) and joining (J) gene segments of each sequence using the IMGT [57] (for V,D,J) and VBASE2 (for V) [58] databases with BLAST [59]. In the case of discrepancy between V segment assignments from the two databases, the assignment leading to the lowest number of somatic mutations is chosen. As cells have at most 20 mutations, all assignments were unambiguous with a V(D)J alignment score over 95%. Next, functional V(D)J sequences are grouped together into clones if they shared the same V, J and D gene segments. The grouped sequences from each clone are then aligned with ClustalW [60]. This second alignment is necessary to infer a mutation matrix, as tree inference algorithms typically require sequences to be aligned and of the same length, while experimentally determined sequences have different lengths, and include insertions and deletions. The corresponding trees are computed with SCITE [61], a stochastic search algorithm based on Markov chain Monte Carlo sampling. The root node is defined by taking the unmutated V(D)J germline and filling the remaining junction region with the consensus sequence of all available sequences. Finally, the obtained trees are analyzed with the BioPhylo Python library [62] to compute the main topological tree properties, such as number of leaves, internal nodes, branches, trunk, etc.

### 2.9. Clonal Diversity

Clonal diversity has been measured using a multi-color fate mapping technology that permanently tags individual B cells and their progeny with different combinations of fluorescent proteins. The method can generate up to 10 different color combinations, thus limiting the experiment to characterising the evolution of a maximum number of 10 clones. A Normalized Dominance Score (NDS), defined as the fraction of all B cells in a GC that carry the dominant color combination, was used to characterize clonal diversity [15,63]. To simulate this experiment, we cluster founders into 10 groups, and define an NDS score as the ratio between the cell count of the dominant cluster over the total number of cells. The NDS is used only for a direct comparison with measurements, but not for other characterisations of the clonal diversity. Later in the paper, we use the clonal dominance (CD), which is simply the ratio between the cell count of a clone cluster and the total number of cells.

### 2.10. Parameter Optimization

Our models include 17 free parameters that need to be optimised. As the amount of quantitative information about the GC dynamics is still limited, we exploit the recent literature to obtain reasonable estimates about the value of some parameters as well as variability bounds (*Lower bound* and *Upper bound* columns in Table 2). Further details can be found in the Appendix A.

In addition, many recent works have experimentally characterized the GC, and we use these data to constrain different properties of the GC response, such as GC kinetics, output cell counts, etc. We further use data from mice experiments when human experiments are not available. We briefly discuss here the constraint bounds on each GC property.

**Number of GC B cells**: A time-resolved study of 3457 GC sections from mice provided information about the GC area $A_{\mathrm{GC}}$ throughout the GC reaction [18]. Assuming that GC are spherical, then $A_{\mathrm{GC}}=\pi \;r_{\mathrm{GC}}^{2}$, from where the radius of the GC, $r_{\mathrm{GC}}$, can be estimated. If we further assume that B cells are the predominant cellular type in the GC, we can estimate the number of B cells in the GC as:   
$$N_{B}\;\mathrm{cell}\approx \frac{V_{\mathrm{GC}}}{V_{B}\;\mathrm{cell}}={\left(\frac{r_{\mathrm{GC}}}{r_{B}\;\mathrm{cell}}\right)}^{3}\;,$$
where we have considered cells to be spherical. Taking $r_{B}\;\mathrm{cell}=6.2\;\mu m$ [64] and $r_{\mathrm{GC}}=80\;\mu m$ (day 9) [18], we get an average of $N_{B}\;\mathrm{cell}\approx 2000$ at the GC peak. As most histological sections do not go cut GCs through the centre, estimations of individual GC volumes are only approximated. Nevertheless, it has been shown that cross-sectional area distributions of spleen sections reflect accurately the broad real-size distributions of GCs [65]. Moreover, as there is significant variability in GC B cell sizes, notably during the cell division cycle [66,67]), the previous number has to be considered only an order of magnitude approximation to the total number of cells. Interestingly, the same study found that GC undergo a fast expansion phase from day 4 to 9, and then progressively decrease in volume by a factor of 2, roughly 10 days after the initial expansion [18].**DZ/LZ ratio**: Despite high variability in size, a typical cellular composition with relatively stable cell ratios of resident T${}_{FHs}$, macrophages, proliferating cells, and apoptotic nuclei seems to be maintained during the established phase of the response [18]. Time-resolved data about the relative size of DZ to LZ is also available from the same study. While the variability in the relative volumes of DZ and LZ is considerably high at all sampled time points (Figure 4B), it stabilizes after the phase of monoclonal expansion to an average of ∼1 during the entire GC reaction [18]. The high variability is consistent with some other studies that found a ratio of 2 [9]. In our simulation, the DZ/LZ ratio is defined as the number of CBs over the number of CCs; however, we only estimate this ratio after day 9 after immunization, hence avoiding the early days of the GC establishment, where the DZ and LZ are still not spatially separated.**Cell death**: Apoptosis is prevalent in the GC, with up to half of all GC B cells dying every 6 h [55]. We use a relatively constant cell death rate of 8%/h in both the DZ and LZ through the entire GC response.**SHM mutations**: Time-resolved data on sequenced BCR mutations, obtained by comparing sequences to the closest known V,D,J germline sequences, suggest that B cell clones undergo roughly 5 mutations every 10 days, and  [6,44]. We use these data to further constrain our model.**GC cellular output**: Finally, we use time-resolve data of the GC cellular output [50] to constrain our model. Because the experiment only measured relative abundances rather than direct cell counts, we scale both the model predictions and experimental measures by the maximum value.

Next, we compute an error for each set of parameters, defined as the sum of the root mean square deviation (RMSD) between the model prediction and experimental results. For a fair measurement of the fitting performances across heterogeneous GCs, the RMSD is normalized by dividing it by the standard deviation of the experimental data averaged over the different time-points—we also average over same time-point replicates, when available, to reduce variability. As no measurement of the cell death standard deviation is available, we arbitrarily assume a standard deviation of 1% cell death/h (measured rate is 8% cell death/h). For each one of the 4 properties for which data is available, i.e., cell count, DZ/LZ ratio, cell death and number of mutations, we compute an independent NRMSD. A NRMSD lower than one indicates that the simulation error is lower than the uncertainties intrinsic to experiments.

The score function is minimized with the maxLIPO algorithm [68], which is both parameter free and provably better than a random search. It is a good alternative to Bayesian optimization methods [69], that typically require the definition of prior assumptions about the function being optimized and thus require domain knowledge. The constraints specified for each parameter, as well as the obtained optimal value, are reported in Table 2. Due to the high computational cost required for parameter optimization with high dimensionality, a simplified ODE system was also used to speed up the maxLIPO search (described in the Appendix A).

## 3. Results

### 3.1. Germinal Center Dynamics

We simulate our GC model using a modified Gillespie algorithm [20] and the parameters optimised as discussed in Section 2.10. Namely, 1000 simulations were run and the normalized root mean square deviation (NRMSE) of each optimised GC property was computed. Results are presented in Figure 4.

Our model is in good agreement with the experimental data, having a fairly low (≲1) NRMSD for all optimised GC properties. For instance, the NRMSD of the GC B cell count is 0.13, indicating a smaller deviation than the standard deviation of the experimentally determined number of GC output cells. A reason for this disparity is the high heterogeneity of GCs in terms of size and cell count [18,65], with measured GCs ranging from ∼500 to ∼10,000 B cells in size. As our model is trained to replicate the mean experimental values, it is expected to show lower standard deviation. Despite this difference, our model recapitulates well the general trends of experimental measurements, e.g., the reduction of GC volume by half in ∼5 days [65], and the shutdown of the GC at around day 40.

Regarding PC/MBC differentiation, the time-dependent GC output of the three models investigated in Section 2.7 reproduces the temporal switch experimentally observed (Figure 4E,F), where memory cell models are produced during the first two weeks of the GC reaction, and plasma cells are produced afterwards [50]. The model based on a fixed threshold has the lowest agreement with experiments, as this model does not lead to a substantial production of MBCs after 2 weeks, while MBCs are produced during the whole GC reaction, albeit at a moderate rate at late time-points. On the other hand, both the dynamic threshold and stochastic differentiation model are compatible with experimental data (average NRMSD = 0.52 and 0.59), with the stochastic model fitting the data slightly better.

To gain insight into which parameters have a higher impact on the GC dynamics, we perform a stability analysis (Appendix A). The analysis reveals that the GC kinetics is most sensitive to changes in the parameters that regulate the selection process by T${}_{FHs}$, such as the amount of T${}_{FHs}$ signals necessary for a LZ B cell to recirculate into the DZ, as well as the amount of cellular divisions it undergoes once there. Similarly, the probability δ of dying in the DZ after a mutation leads to a nonproductive receptor critically influences the number of cells in the GC.

We also perform a sensitivity analysis to quantify the effect of changes in the parameters in the simulation output. Due to the complexity of the model, we only consider individual changes in parameters, namely, we explore changes of ±10% of each parameter, while keeping the rest of the parameters unmodified [24]. The changes in the model output are quantified by measuring the change in NRMSE${}_{\mathrm{total}}$ over all quantified GC properties (Table 2). Some parameter changes mainly affect the GC kinetics, e.g., the GC B cell counts and the DZ/LZ ratio, while others have an impact on the affinity maturation process, e.g., clonal diversity and number of acquired mutations. As an example, the lethal mutation probability δ has a strong impact on the GC kinetics, as a 10% decrease leads to a 660% change in the total NRMSD. On the other hand, the silent mutation probability σ has a low impact, as a 10% decrease only leads to a change in the total NRMSD of <10% (Appendix A).

### 3.2. Comparison of Simulated and Experimentally Determined BCR Sequences

A direct comparison of sequences between sequenced GC B cell repertoires and sequences inferred with our model is not possible due to our abstract representation of the variable region of the BCR. As an alternative, we use phylogenetic trees to visualize the somatic evolution of B cells during the GC response. To construct experimental trees, we use single-cell BCR sequencing data obtained from individual GCs [15]. Specifically, the sequences of the Ig heavy chain of ∼1500 single B cells from 18 distinct GCs—with an average of roughly 80 cells per GC—were used to build 266 trees containing at least 2 cells. All measurements were performed on mice at day 15 after immunization with different antigens. By sampling midway through the GC reaction, intermediate BCRs that might have populated the GC at earlier time-points are lost, and the inference sometimes shows long trunks, representing the pruned evolutionary process of the sampled cells, in agreement with earlier observations [29]. Co-existing clones are only seen close to the tree leaves (Figure 5A), where selection is currently taking place. Regarding the inference from simulated data, trees can be straightforwardly reconstructed, as the whole lineage is available in a simulation. To reproduce the experimental sampling conditions with the simulated data, we uniformly sampled 80 cells at day 15 after immunization from our model.

The experimental and simulated trees were analyzed with the BioPhylo python library [62] to compute important topological properties, such as number of leaves, average leaf depth, and trunk length. The distributions of these properties in both the simulated and the experimental tress are shown in Figure 5B–D. Simulated distributions are obtained after averaging 1000 runs in our model. Overall, the distributions obtained from simulated and experimental data are in good agreement (S > 0.6, Figure 5). However, experiments suggest that the density function of cell mutations per tree is broader than what our model predicts. This could be explained by the fact that all mutations have equal contributions in our model, where one mutation can only increment the affinity by $\frac{1}{N_{\mathrm{CDR}}}$, with $N_{\mathrm{CDR}}=25$. Such limitation prevents cells with a low number of mutations from having a high-affinity BCR, thus constraining the simulated density function to be narrower. Another contributing factor could be the fact that our simulation only considers the CDR region, and does not keep track of silent mutations that might occur in the FWR region.

### 3.3. Visualizing Affinity Maturation

Figure 6A depicts the evolution of the average affinity in the GC through its lifespan. Noticeably, affinity does not substantially improve during the first days of the GC reaction, corresponding to the phase of monoclonal expansion where B cells divide without selection pressure. After day 10, affinity increases linearly throughout most of the lifespan of the GC, only showing slower improvement rates near the end of the GC reaction, where the remaining B cells have presumably developed good receptors against the exposed antigens, and new mutations are likely to be detrimental to the overall affinity.

For visualization purposes, we randomly sampled two cells every day from the dominant clone, where dominance was established by analyzing clonal cell counts at the end of the simulation, and computed a tree from the obtained set of BCR sequences (Figure 6B). Since we sample uniformly throughout the lifespan of the GC, we do not observe the long trunk found in experimentally inferred trees due to pruned clones. Although the specific structure of the tree shows high variability as a consequence of the strong undersamplig, the tree clearly shows how affinity maturates as a result of clonal evolution. Noticeable also is the clear correlation between increased affinity and number of SHMs (the number of mutations is equal to the number of nodes connecting the root cell and a leaf). Resampling the BCR set multiple times did not significantly change the properties of the tree.

### 3.4. Visualizing Clonal Competition and Clonal Burst

To better understand the complex process of clonal competition and the mechanisms behind clonal bursts observed in a subset of GCs [15], we plot the evolution of the number of clonal families in the GC as a function of time (Figure 7A), where clonal families are defined in our simulations as the cells descending from the same founder cell. Our model reveals that the number of clones increases until day 8, corresponding to the continuous entry of new founder cells in the GC. Then, shortly after the monoclonal expansion phase, the GC experiences a rapid loss of clonal diversity due to the high selection pressure exerted in the LZ on the still immature CCs, followed by a more moderate decrease that lasts until the end of the GC reaction. These results are consistent with experimental characterizations that have revealed that clonal diversity is relatively high during the early days of the GC, followed by a phase lasting several weeks where a few clones quickly dominate the reaction [15].

While the evolution of the number of clones in the GC has not been measured quantitatively, clonal diversity has been characterized using a Normalized Dominance Score (NDS) [15] (Section 2.9). An interesting finding of this study is that, while most GCs retain substantial clonal diversity within the 3-week lifetime of the GC response, rapid and massive expansion of higher-affinity SHM variants (clonal bursts) leads to substantial loss of diversity in a subset of GCs. As illustrated in Figure 7B, our model quantitatively reproduces the measured NDS with a low NRMSD (0.40). Both the simulations and the experimental characterizations reveal a high variance in the NDS score. For instance, our model predicts that 2.5% of GCs reach clonal dominance (NDS = 1) at day 30, and 25% at day 40. This is expected if stochastic effects play a fundamental role in the emergence of dominance.

To investigate the emergence of clonal dominance in more detail, we plot the Clonal Dominance (CD), which clusters cells that descend from the same founder cell (Section 2.9), of individual families in two representative simulations in Figure 8A,C. The first simulation (1) shows a GC with no observable clonal bursts. The CD of the dominant clone (green line) only becomes dominant after day 15, but other competing clones are still significantly present until much later (∼30 days). In the second simulation (2), the dominant clones (brown line) outcompete other clones as early as day 10, with most of the co-existing clones disappearing by day 25. It is interesting to notice that in the latter scenario, representing a GC that underwent a clonal burst, the winning clone acquired an early advantage (before day 10) in terms of affinity (Figure 8D), which enabled the clone to proliferate and gain dominance. Although other clones reached higher levels of affinity at later time points, e.g., from days 15 to 25, the early advantage that led to a higher proliferation rate was sufficient to ensure dominance throughout the whole GC reaction.

The simulation shown in Figure 8 (2) seems to suggest that there is a time delay of roughly 7 days between the initial clonal affinity advantage and the clonal burst induced by it, corresponding to multiple LZ-DZ cell cycles—in our model, a full DZ–LZ cycle lasts ∼1–2 days depending on the number of divisions in the DZ. To better quantify the delay between the increase in affinity and the appearance of clonal dominance, we performed a Time Lagged Cross Correlation analysis [70] between the clonal affinity and the clonal dominance time series of each clone. Briefly, the analysis computes the Pearson correlation of one time series with the respect to the second one shifted by a time offset. By trying different values for the offset, one can find the optimal time delay that maximises the correlation. We collected time series data from 1000 clone dominance-affinity pairs by taking the 10 most abundant clones of 100 GC simulations. The results obtained, highlighted in Figure 9, confirm that the correlation is maximized for a time delay of 7.1 days, which suggests a causal relationship between a random increase in affinity and the emergent appearance of dominance one week later. A delay of 7 days corresponds to roughly 3–4 rounds of DZ–LZ B cell cycles. This delay is consistent with the expectation that the production of high-affinity antibodies requires several rounds of cell division and affinity maturation. More specifically, with an average of three divisions per DZ–LZ cycle, our model suggests that one round of somatic hyperpermutation is not enough to achieve clonal dominance.

## 4. Discussion

We have presented a stochastic multiscale model of the GC that combines an abstract representation of individual B cell receptors (BCRs) as strings of nucleotides, a deterministic regulatory representation of the transcriptional program associated with B cell differentiation [20] and a stochastic representation of the cellular interactions that shape the GC reaction. The model parameters were optimised using time-resolved data from various experimental characterizations of the GC kinetics and GC clonal diversity [15,18,44,50,55]. Overall, our model recapitulates the GC dynamics characterized experimentally, including the GC B cell counts at different times of the GC reaction, the DZ/LZ volumetric ratio, the cell death and accumulation of mutations as a result of SHM. Compared to existing models, our model accounts for a broader range of important biological processes, e.g., centroblast apoptosis, a more realistic description of antigen capture, and faithfully replicates experimental data. A sensitivity analysis reveals that the GC kinetics is very sensitive to small changes in some key parameters, indicating that the B cell maturation is a tightly regulated program. In particular, slight differences in the T${}_{FH}$ selection process have a high impact in the GC dynamics, and can potentially lead to a drastic increase in both the number of dividing centroblasts as well as differentiating cells. Such sensitivity could explain the high variability observed in the GC sizes [18]. Our optimisation approach enables the easy integration of new experimental data that might become available in the near future.

With the help of our model, we computationally investigated three candidate mechanisms that might potentially govern the terminal differentiation of GC B cells into PCs or MBCs. Namely, (i) a deterministic scenario, where the amount of captured antigen determines cell fate; (ii) a dynamic model, similar to the deterministic model, but where the threshold of captured antigen to become a PC increases with the lifespan of the GC; and (iii) a probabilistic scenario, where the amount of exposed antigen determines the cellular outcome in a probabilistic fashion, i.e., cells with high and low amounts of antigen have a non-zero probability of becoming either a PC or a MBC. Although the three models perform reasonably well and reproduce the observed temporal switch from memory cells to plasma cell production accurately [50], the stochastic differentiation model has the best agreement with experimental data (Figure 4E,F). One important point is that while the three models differ in the absolute counts of PCs and MBCs produced (Appendix A), the experimental data only reports relative changes across time-points, which might hide global changes in cellular output. Additional experimental data on the GC output will help further differentiate between the 3 considered scenarios.

The amount of T${}_{FHs}$ plays a critical role in determining the GC outcome. Models where the number of T${}_{FHs}$ is fixed at a constant value tend to be stable, with the GC reaching a steady state upon which it does not decreases or increases in size, contrary to observations (Figure 4A). We therefore assume that T${}_{FHs}$ can enter and leave the GC during the whole reaction [40], which leads to having an approximately constant density of T${}_{FHs}$ [18]. With this assumption, the sizes of our simulated GCs closely resemble experimental determinations, and the GCs naturally become extinguished by day ∼40 (Figure 4A). Critical to achieving this self-shutdown is the choice of parameters. Indeed, a very rough estimation of the rate of variation of the total number of GB B cells, ${\dot{N}}_{B}$, shows that this number approximately evolves according to ${\dot{N}}_{B}⋍0.6\;\cdot \;r_{\mathrm{division}}\;\cdot \;N_{\mathrm{CB}}-r_{\mathrm{apoptosis}}\;\cdot \;N_{\mathrm{CC}}-N_{{\mathrm{CC}}_{\mathrm{sel}}}\;\cdot \;r_{\mathrm{exit}}$, where 0.6 is a correction factor that accounts for apoptotic and non-dividing centroblasts. Taking $N_{\mathrm{CC}}/N_{{\mathrm{CC}}_{\mathrm{sel}}}⋍N_{\mathrm{CB}}/N_{{\mathrm{CC}}_{\mathrm{sel}}}⋍100$, as predicted by our model, our choice of parameters leads to ${\dot{N}}_{B}⋍-0.4\;N_{{\mathrm{CC}}_{\mathrm{sel}}}<0$, explaining the observed reduction in $N_{B}$ after day 9 (a detailed ODE analysis can be found in the Appendix A). Hence, after the initial phase of monoclonal expansion, our simulated GCs enter a phase of slow decrease in size until they naturally shutdown around day 40. Of course, additional mechanisms may be at play in terminating the GC reaction, such as antibody feedback [71]. Similarly, asynchronous onset and intercommunication between GCs might alter the efficiency of antibody response and lead to compromised efficiency and earlier termination of late initialized GCs [21]. Future renditions of our work will explicitly investigate the mechanisms by which affinity influences antigen uptake and antigen presentation, including its contribution to the termination of the GC reaction.

We also generated BCR sequences from our simulations and compare the simulated sequences with single-cell BCR sequencing data from individual GCs by means of phylogenetic trees. Each tree represents the evolutionary process that took place in the GC and the acquisition of new mutations that conferred higher affinity to the most fitting clones. The comparison of important tree topological properties, such as the number of leaves, the average depth, or the length of the trunk, reveals good agreement between our model and experiments. We find our predicted distribution of mutations per cell to be narrower than experimentally characterized (Figure 5), which could indicate that our current model of affinity, where one mutation can only increment the affinity by $\frac{1}{N_{\mathrm{CDR}}}$, constrains the density function of the mutations count to be narrower. Furthermore, because of our explicit modelling of the BCR sequences, our model enables the investigation of the clonal diversity within the GC and the emergence of dominance. A first observation is that B cells with the same affinity can have very different outcomes, resulting in a high variability of the evolution of clonal diversity across simulations, which matches experimental observations [15] (Figure 7). Interestingly, a deeper investigation of the emergence of clonal dominance demonstrates that small advantages in the affinity to antigen acquired through random mutations are amplified and result in clonal dominance within a week (Figure 8 and Figure 9). This delayed correlation might explain why previous computational models found limited correlation between clone abundance and affinity throughout the GC lifespan [24].

As future work, we would like to investigate approaches to overcome technical limitations of the Gillespie algorithm, such as the assumption that random interactions can be modelled as a memoryless Poissonian process. This hypothesis might not be accurate for some biological programs, such as apoptosis or cell division, which tend to be better represented using log-normal distributions [72,73]. We also plan to explore generalizations of the Gillespie algorithm that incorporate both individual properties and non-Poissonian processes, while still being statistically correct.

Our model does not include regulation by T follicular regulatory, T${}_{FR}$, cells. T${}_{FR}$ cells have been implicated in the negative regulation of GC B cell activation, affinity maturation and the differentiation of plasma cells [74]. Furthermore, T${}_{FRs}$ may have a role in the negative selection of autoreactive B cells that make up a significant part of GC B cells (8% according to [55]). Indeed, ex vivo TCR repertoire analysis shows that T${}_{FHs}$ and T${}_{FRs}$ cells exhibit different TCR repertoires, with the latter more closely resembling that of regulatory T cells and being skewed toward self-antigens [75]. This finding leads to the hypothesis that T${}_{FRs}$ might negatively regulate potentially autoreactive B cells to suppress autoimmunity [76]. Further work will integrate T${}_{FRs}$ and autoreactive B cells to our model, and will provide more insight behind the mechanism of autoimmunity.

Finally, an improved model to compute BCR affinity that integrates information about BCR sequences and about the structural shape of both the antibody variable region and the antigen can significantly enhance our knowledge of the mechanisms involved in the adaptive immune response [45]. Attempts in this direction have already been made, for instance by representing BCRs as a chain of 10 amino acids on a rigid lattice [28]. Despite these promising works, a full integration of the 3D structure of both the BCR and the antigen remains challenging. An alternative approach is to focus on predicting binding affinity directly from sequence and transcriptomic data. This is already routinely performed to predict drug sensitivity, where the binding affinity between a drug compound and a protein is predicted using transcriptomic data and chemical information without explicitly accounting for the 3D structure of either molecule [77,78]. An advantage of such approaches is that they enable the in silico engineering of new compounds with improved biochemical properties [79]. The adaption of these models to the task of predicting the binding affinity between a receptor and an antigen or therapeutic compound, e.g., a vaccine, would open exciting opportunities for synthetic antibody engineering or vaccines in silico optimisation. Work in this direction could have vast repercussions in understanding the emergence of broadly neutralizing antibodies, designing serological tests or vaccines against mutating pathogens, such as SARS-CoV-2.

## Figures and Tables

**Figure 1 cells-09-01448-f001:**
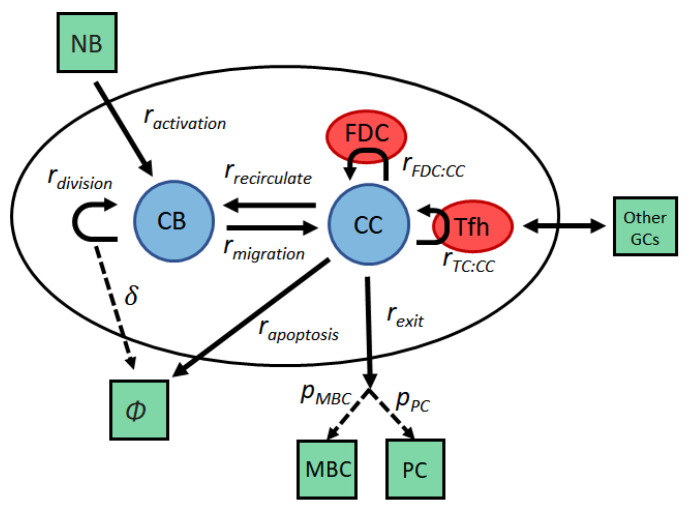
Probabilistic model for the germinal center (GC) dynamics. Activated naive B cells (NB) enter the GC as centroblasts (CBs), which rapidly divide and mutate their immunoglobulin variable regions. If mutation renders a BCR nonproductive, the CB commits apoptosis (∅). After a few hours, CBs migrate to the light zone and become centrocytes (CCs). High-affinity CCs can acquire large amounts of antigen from follicular dendritic cells (FDCs), which makes them more likely to receive survival signals from helper T cells (T${}_{FH}$). Low affinity CCs, which acquire little or no antigen, cannot compete for T cell help and die through apoptosis (∅). CCs that have received sufficient T${}_{FH}$ signals may either recirculate into the dark zone for another round of division and hypermutation, or leave the GC as a memory B cell (MBC) or a plasma cell (PC).

**Figure 2 cells-09-01448-f002:**
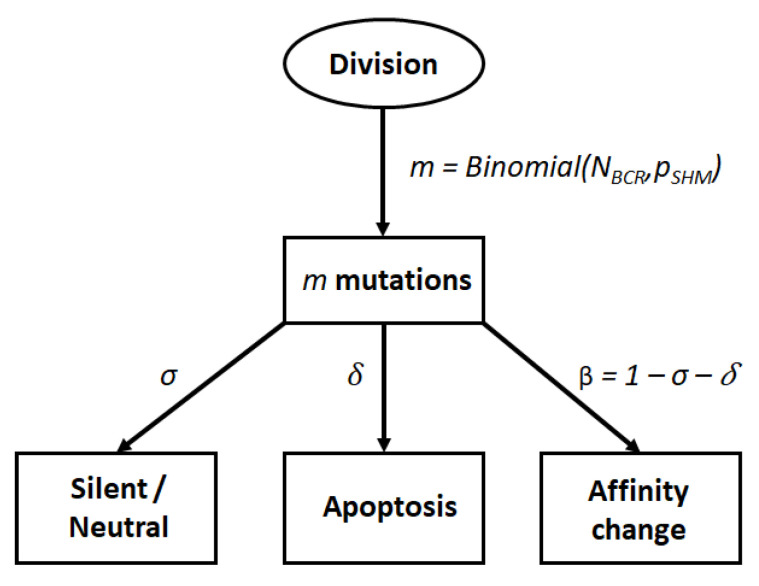
Decision tree to model cell fate after somatic hypermutation (SHM). The number of mutations in the BCR after each centroblast division is sampled from a binomial distribution. Each mutation can produce a BCR of similar affinity, lead to cell apoptosis or produce a BCR of different affinity with probability σ, δ, and β respectively.

**Figure 3 cells-09-01448-f003:**
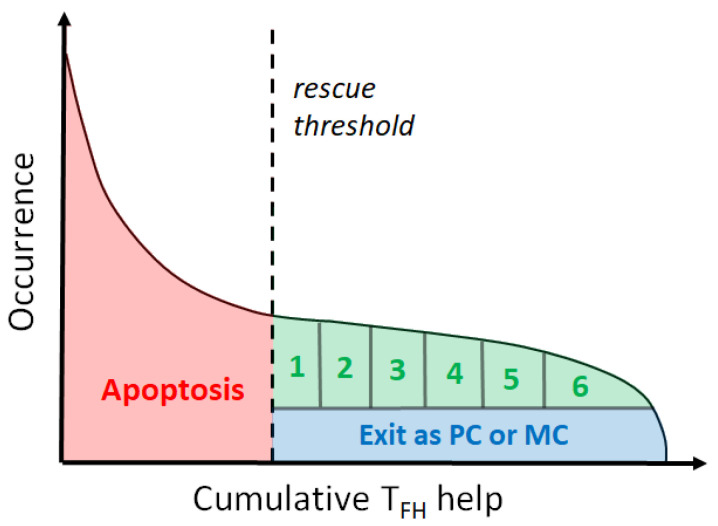
Fate of B cells after interaction with T${}_{\mathrm{FH}}$. Cells that do not receive sufficient help from T${}_{\mathrm{FHs}}$ undergo apoptosis. The remaining cells either leave the GC and differentiate into a PC or MBC, or cycle back to the dark zone, where they undergo up to 6 divisions depending on the amount of T${}_{\mathrm{FH}}$ signals received.

**Figure 4 cells-09-01448-f004:**
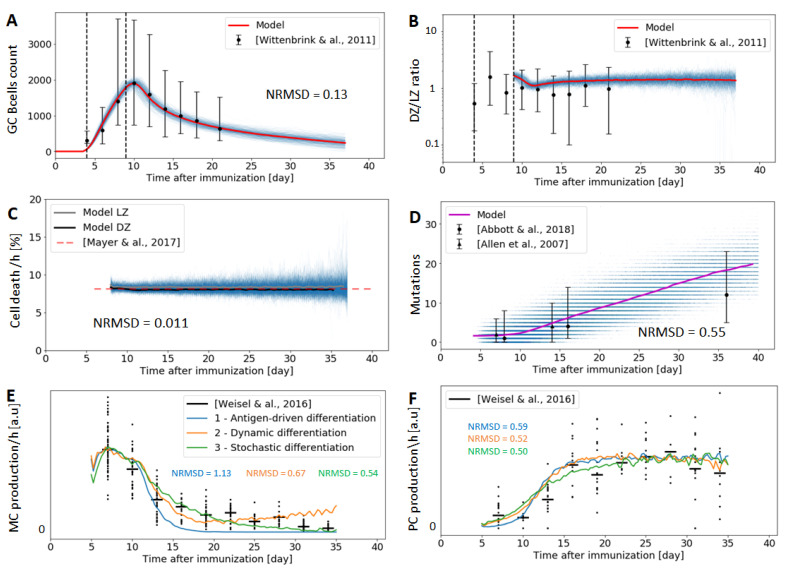
Experimental data vs. our model after 1000 simulations, along with the obtained normalized root mean square deviation (NRMSE) of each measured GC property. Thin blue lines represent one single Gillespie simulation, while thick lines are the average over all simulations. Experimental results are represented by a tick bar corresponding to the median and 1st-3rd quartiles, the central point being the median. Figures (**A**,**B**) depict simulations and experimental data about cell count and DZ/LZ volumetric ratio respectively, the latter derived from [18]. Figure (**C**) shows the death rate percentages of B cells in both the DZ and LZ (8%/h) [55]. Figure (**D**) shows the mutation rates at different time points, and roughly corresponds to 5 mutations every 10 days [6,44]. Figures (**E**,**F**) illustrate the temporal switch from memory B cell to plasma cell production simulated with our model using 3 different differentiation scenarios and experimentally measured [50].

**Figure 5 cells-09-01448-f005:**
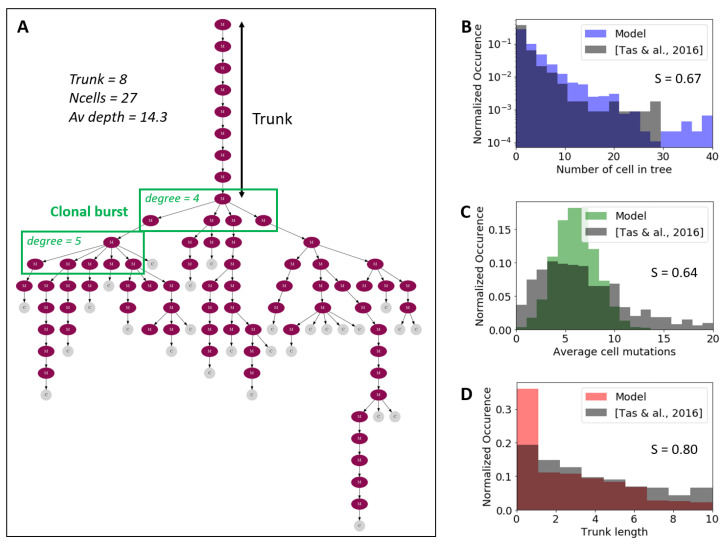
Phylogentic trees from BCR sequences at day 15 after immunization. (**A**) Representative phylogenetic tree inferred from single-cell BCR sequencing extracted from individual GCs [15]. The purple nodes (M) represent mutation and the gray node (C) represent individual cells. (**B**–**D**) distribution of tree topological properties (number of leaves, average leaf depth and trunk length) in experimental and simulated trees. The score S is defined as Overlapping Area/Joint Area of the histograms.

**Figure 6 cells-09-01448-f006:**
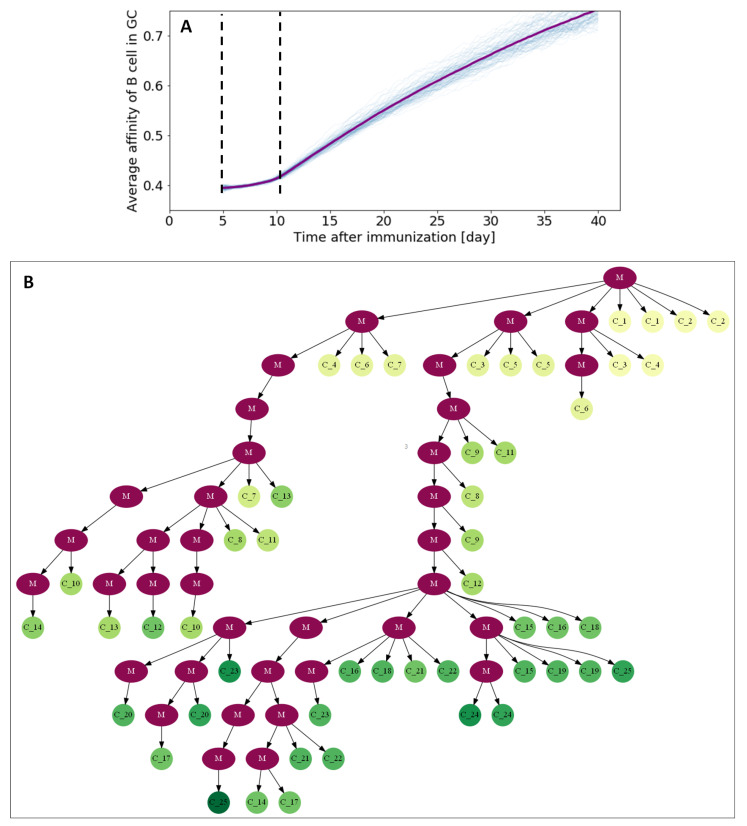
Affinity maturation in GC through its lifetime. (**A**) Mean BCR affinity as a function of time. (**B**) Tree representation of affinity maturation. Two cells were sampled every day for 25 days (50 cells in total). The purple nodes (M) represent mutation, and the color of cell nodes represents the BCR affinity (yellow = low and green = high), the cell label subscript is the day on which the cell was sampled.

**Figure 7 cells-09-01448-f007:**
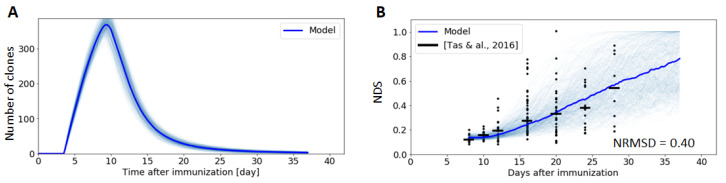
Clonal diversity evolution during the GC reaction. Thin blue lines represent one single Gillespie simulation, while thick lines are the average of 1000 simulations. Figure (**A**) depicts the B cell clonal evolution through the GC response, and (**B**) the Normalized Dominance Score (NDS), i.e., the fraction of B cells in the dominant clone, as reported by [15]. Experimental results are represented by a scatter plot, where individual points represent single measurements and the bar indicates the median.

**Figure 8 cells-09-01448-f008:**
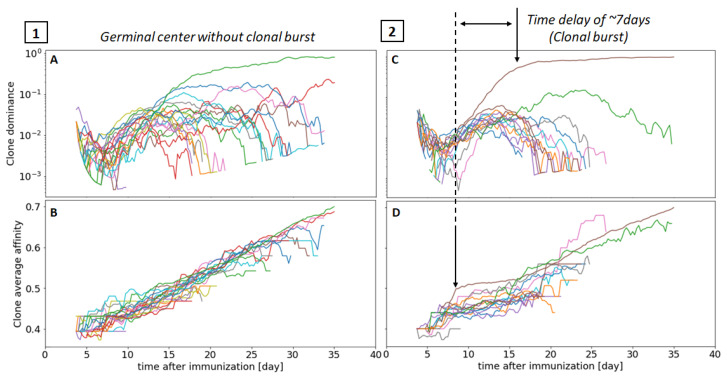
Clonal dominance and clonal affinity maturation for two representative GC simulations, without (1) and with (2) clonal burst. Figures (**A**,**C**) showcase the clonal dominance score evolution in the GC, and (**B**,**D**), the average affinity of each clonal family as a function of time. Only the most abundant 50 clones are shown to simplify the presentation.

**Figure 9 cells-09-01448-f009:**
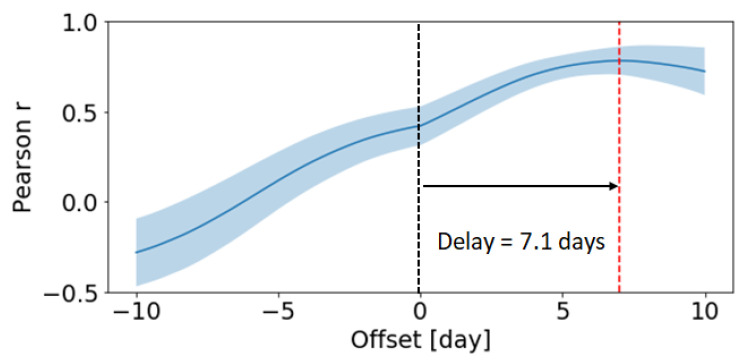
Time Lagged Cross Correlation analysis between clonal dominance and clonal affinity time series. The Pearson correlation is averaged from the 10 most abundant clones of 100 germinal center simulations (resulting in 1000 clone dominance-affinity pairs). The shaded region corresponds to one standard deviation. The correlation peaks at 0.79 at an offset of 7.1 days.

**Table 1 cells-09-01448-t001:** Set of reactions used in our germinal center model, along with the reactions rate values obtained after optimization, as detailed in Section 2.10.

Reaction	Description	Parameter Value
$\mathrm{NB}\overset{r_{\mathrm{activation}}}{\to }\mathrm{CB}$	B cell activation	$r_{\mathrm{activation}}=3.94\;/h$
$\mathrm{CB}\overset{r_{\mathrm{division}}}{\to }2\mathrm{CB}$ where each daughter CB undergoes apoptosis ($\mathrm{CB}\overset{}{\to }\emptyset$) with probability δ per acquired mutation.	Centroblast division	$r_{\mathrm{division}}=0.134\;/h$ δ=0.52
$\mathrm{CB}\overset{r_{\mathrm{migration}}}{\to }\mathrm{CC}$	Centroblast migration to LZ	$r_{\mathrm{migration}}=3.75\;/h$
$\mathrm{CC}\overset{r_{\mathrm{apoptosis}}}{\to }\emptyset$	Centrocyte apoptosis	$r_{\mathrm{apoptosis}}=0.084\;/h$
$\mathrm{CC}+\mathrm{FDC}\overset{r_{\mathrm{FDC}\text{:}\mathrm{CC}}}{\to }\mathrm{CC}\left(\mathrm{pMHC}\right)+\mathrm{FDC}$	Centrocyte antigen uptake	$r_{\mathrm{FDC}\text{:}\mathrm{CC}}=\frac{40}{N_{\mathrm{CC}}}\;/h$
$\mathrm{CC}+T{}_{FH}\overset{r_{\mathrm{TC}\text{:}\mathrm{CC}}}{\to }\left[\mathrm{CCTC}\right]$	Centrocyte binding to T${}_{FH}$	$r_{\mathrm{TC}\text{:}\mathrm{CC}}=\frac{145}{N_{\mathrm{CC}}}\;/h$
$\left[{\mathrm{CC}}_{1}\mathrm{TC}\right]+{\mathrm{CC}}_{2}\overset{r_{\mathrm{TC}\text{:}\mathrm{CC}}}{\to }{\mathrm{CC}}_{1}+\left[{\mathrm{CC}}_{2}\mathrm{TC}\right]$	Centrocyte T${}_{FH}$ switch	
$\left[\mathrm{CCTC}\right]\overset{r_{\mathrm{unbinding}}}{\to }{\mathrm{CC}}_{\mathrm{sel}}+T{}_{FH}$	Centrocyte spontaneous unbinding	$r_{\mathrm{division}}=2\;/h$
${\mathrm{CC}}_{\mathrm{sel}}\overset{r_{\mathrm{recirculation}}}{\to }\mathrm{CB}$	Centrocyte recirculation to DZ	$r_{\mathrm{recirculate}}=3.75\;/h$
${\mathrm{CC}}_{\mathrm{sel}}\overset{r_{\mathrm{exit}}}{\to }\mathrm{MBC}\;\mathrm{or}\;\mathrm{PC}$ with probability $p_{\mathrm{MBC}}:\mathrm{CC}\overset{}{\to }\mathrm{MBC}$, with probability $p_{\mathrm{PC}}:\mathrm{CC}\overset{}{\to }\mathrm{PC}$.	Centrocyte exit	$r_{\mathrm{exit}}=1.6\;/h$ see Section 2.7 for descriptions of $p_{\mathrm{MBC}}$ and $p_{\mathrm{PC}}$

**Table 2 cells-09-01448-t002:** Parameter bounds and optimized values used in our model. The *Lower bound* and *Upper bound* are derived from the literature; the fitted values are obtained with the maxLIPO optimization algorithm. The *Sensitivity* column indicates how sensitive the score function is to a ±10% change in the parameter value. The labels low, moderate, high correspond respectively to a <10%, <100% and <1000% change of the score function. Further details can be found in the Appendix A.

	Parameter	Description	Lower Bound	Upper Bound	Fitted	Sensitivity	Mainly Affects
Intercellular	$r_{\mathrm{activation}}$ [$h^{-1}$]	B cell activation rate	1	10	3.94	high	Value of GC peak
	$r_{\mathrm{division}}$ [$h^{-1}$]	CB division rate	0.08	0.16	0.134	moderate	DZ/LZ ratio
	$r_{\mathrm{migration}}$ [$h^{-1}$]	CB migration rate	0.15	3.75	3.75	low	DZ/LZ ratio
	$r_{\mathrm{apoptosis}}$ [$h^{-1}$]	CC apoptosis rate	0.06	0.16	0.084	high	GC decay
	$r_{\mathrm{exit}}$ [$h^{-1}$]	CC${}_{\mathrm{sel}}$ exit rate	0.41	3.75	1.6	high	GC decay
	$r_{\mathrm{recirculate}}$ [$h^{-1}$]	CC${}_{\mathrm{sel}}$ recirculation rate	fixed to 3.75			high	GC decay
	$r_{\mathrm{FDC}\text{:}\mathrm{CC}}$ [$h^{-1}$]	FDC encounter rate	fixed to $40/N_{\mathrm{CC}}$			low	Clonal competition
	$r_{\mathrm{TC}\text{:}\mathrm{CC}}$ [$h^{-1}$]	T${}_{FH}$ encounter rate	fixed to $145/N_{\mathrm{CC}}$			moderate	Clonal competition
	$r_{\mathrm{unbinding}}$ [$h^{-1}$]	T${}_{FH}$ unbinding rate	fixed to 2			high	GC decay
	$\alpha_{\mathrm{TC}}=N_{\mathrm{TC}}/N_{\mathrm{CC}}$	T${}_{FH}$ to CC ratio	1/100	1/7	1/46	high	GC decay
	$N_{\mathrm{FDC}}$	Number of FDCs	fixed to 250			low	Clonal competition
Intracellular	δ	Lethal mutation probability	0.1	0.9	0.52	high	GC decay
	σ	Silent mutation probability	fixed to 0.28			low	Affinity maturation
	$p_{\mathrm{SHM}}$	Mutation rate	fixed to $1\times 10^{-3}$			high	GC decay
	$N_{\mathrm{CDR}}$	CDR length (nucleotides)	fixed to 25			low	Affinity maturation
PC/MBC	${\mathrm{pMHC}}_{\mathrm{threshold}}$	Model 1 parameter	0	1	0.46	moderate	PC/MBC production
	$t_{0}$	Model 2 parameter	0	*∞*	3 days	moderate	PC/MBC production
	$a_{0}$	Model 2 parameter	0	1	0.05	moderate	PC/MBC production
	*k*	Model 3 parameter	0	*∞*	14	moderate	PC/MBC production
	$p_{0}$	Model 3 parameter	0	1	0.4	moderate	PC/MBC production

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
