# Peer review of "Computational Model Reveals a Stochastic Mechanism behind Germinal Center Clonal Bursts"

_cells, 2020, doi:10.3390/cells9061448_

Round 1

Reviewer 1 Report

Herein the authors described the lifespan of germinal centers (GC) and interested in the ways B cells can proliferate, differentiate, and mutate their antibody genes in response to the presence of foreign antigens. They used numerous literature data and amounts of knowledge to parameterize their mathematical model. This is an enormous/impressive work and a very detailed introduction to GC lifespan. I took large pleasure to read it and this is of great interest. 

The authors found that stochastic effects probably play a fundamental role in the emergence of dominance.

My main comments concern the 'discussion section'.

1° ) The authors should  detail the clinical applications of their model and not only cite 'vaccines against SARS-Cov-2' as a marketing argument within the conclusion and 'autoimmune disease and GC derived lymphomas'. I really would like to know and understand how it would be useful. Please develop these points

2°) How to integrate the complexity of the 3D structure of an antigen in this model ? So, how to design a vaccine without integrating this information ?

3°) HCV-induced lymphomas (cryoglobulinemia) would be good examples, as well as Sjögren's Syndrome.

Author Response

We highly appreciate the mostly positive comments about our work. Regarding the specific comments:

  1. We apologize for only mentioning COVID without further developing the main idea. Some of us are currently working on the development of generative models using deep learning for drug design. We are now investigating how to adapt some of the developed technologies to model the binding of a therapeutic agent (such as a vaccine) to a B cell receptor. This is work in progress, but the main idea is to expand the current GC model by adding a new module that predicts the binding between a BCR and a candidate vaccine using deep learning approaches. If such adaptation can be done with a reasonable computational cost, the new model could be used to simulate the maturation of B cells after immunization. We have now added a short description of these ideas in the discussion.

  1. Modelling the complexity of the 3D structure of the receptor is indeed challenging. Recently, however, deep learning is demonstrating promising capabilities to predict affinity binding without the need of running traditional approaches based on molecular dynamic simulations. The main idea here is that a deep learning network, if trained on sufficient data, can learn to predict binding affinity directly from the sequence of amino acids. The senior author has worked in this idea in the context of anti-cancer compound development. This work is under review in Nature Communications (a preprint is available at https://arxiv.org/abs/1909.05114). The idea we are currently investigating is whether such approaches can be applied to modelling the interaction between a compound (e.g. a vaccine) and a BCR, where BCRs are modeled using the sequence, rather than the 3D structure. This is now briefly addressed in the discussion.

  1. Cancers of the immune system (e.g. HCV-induced lymphomas and cryoglobulinemia) and autoimmune diseases (e.g. Sjögren's Syndrome) are indeed good exemplary proofs of concepts to demonstrate the capabilities of an improved model of the GC. We thank the reviewer for the suggestions and will keep these diseases in mind for future work.

Reviewer 2 Report

This is an interesting paper presenting a new in silico model of affinity maturation in the germinal centre. No model is perfect, which is one of the major benefits of computer modelling. As soon as the model does not fit experimental data, it is time to improve the in silico model which may lead to the discovery of new rules. The current paper attempts to include a new way to code BCR affinity in the model, which is computationally affordable, but to somewhat more realistically mirrors biology. This is relevant for the development not only of this model, but for models of GC responses in general. Three variants of the model are presented. These are not variants of BCR affinity modelling, but how BCR affinity determines B cell selection.

Main questions are:

  •  The reason a 4 character code was used, was to replicate the genetic code. Why has sigma (silent mutation probability) set to 0.25? Looking at the real genetic code, this should be 0.15. I also can’t find  of 0.25 in the literature cited (suppl. Ref 11)
  • A main concern is how GC volume was calculated. Histological tissue cutting generates random sections through GCs. Therefore the area of the GC AGC is not pir2GC. This is only correct if the section is through the centre of the GC. The average volume of all GCs on a tissue can be estimated by calculating the percentage area of GC area of overall tissue section area [1]. Most sections through GCs however will not go through the centre and an estimate of volume of individual GC is therefore not possible. Estimating volumes of individual GCs can only be done by reconstructing individual GC from serial sections (ref 18). I suspect this the reason for the huge variability of GC B cells counts in Fig. 4A. These calculations or estimates should be redone.
  • - B cell development in the GC is about selecting for affinity. Therefore, a central problem about modelling the GC response is how BCR affinity is translated into B cell selection. The authors do a good effort on modelling affinity, and the selection of B cells by T cells according to antigen presentation intensity is how we think the process works. However, there is not much thought how affinity translates into antigen presenting capacity. Line 147 states that CC acquire antigen in an affinity dependent manner. There is not much evidence that affinity by itself determines how well a B cell takes up antigen. Future models should put some thoughts into how affinity translates into antigen uptake and antigen presentation, e.g. by antibody feedback (ref 68).
  • - It is somewhat disappointing that not more was done to compare models 1-3 of B cell selection. All I can see are the two parameters in Fig. 4E and F.
  • - The delay between onset of affinity maturation and appearance of higher affinity antibody (5 d) seems very long. Is this realistic?
  • References: 1. Weibel, E.R., Principles and methods for the morphometric study of the lung and other organs. Laboratory Investigation, 1963. 12(1): p. 131-155

Author Response

  1. Sigma includes both the silent mutation and neutral mutation probability. A silent mutation refers to a nucleotide change that does not affect the amino acid sequence (0.15 as the reviewer mentions). A neutral mutation refers to an amino acid change that does not change the affinity of the BCR, e.g. a mutation in the FWR region.

The Figure 2 of [1] (Fate of each somatic hypermutation) indicates a probability for neutral mutations: sigma = 0.75*0.75*0.5 = 0.28 (probability of mutation being in the FWR region * probability of AA being replaced * probability of the mutation being not lethal). It is indeed not 0.25 and the value in the paper has been updated. However, we would like to add that the value of sigma has only a moderate effect on the simulation, as the column 'Sensitivity' in Table 2 of the main text indicates.

[1] Reshetova, Polina, et al. "Computational model reveals limited correlation between germinal center B-cell subclone abundancy and affinity: implications for repertoire sequencing." Frontiers in immunology 8 (2017): 221.

  1. The reviewer is correct, and we agree that estimating GC volumes from histological tissue cutting sections needs to be done carefully to avoid the randomness due to not cutting through the GC center. However, we would like to point out that Wittenbrick & al. estimated the GC volume from seven serial sections (s01–s07), and shown that the cross-sectional area distributions is similar to the volume distribution. The authors concluded that “cross-sectional area distributions of spleen sections reflect broad real-size distributions of GCs.” In particular, the high variance observed in the sections cannot be explained by the randomness of histological tissue cutting alone. We have added a comment to reflect this point.

We would also like to point out that the values we provide in section ‘2.10. Parameter optimization’ are only meant to be rough estimates, as our main concern is the estimation of the GC dynamics, which is to a certain extent relatively independent of individual GC sizes (within a physiological range, of course).

Ref: Wittenbrink N, Weber TS, Klein A, Weiser AA, Zuschratter W, Sibila M, Schuchhardt J, Or-Guil M. Broad volume distributions indicate nonsynchronized growth and suggest sudden collapses of germinal center B cell populations. The journal of immunology. 2010 Feb 1;184(3):1339-47.

  1. Regarding the evidence that affinity by itself determines how well a B cell takes up antigen, while not shown in vivo, as correctly stated by the reviewer, our model follows a widely accepted hypothesis from the work of Victora et al., Cell 143:592 2010. Briefly, using an experimental system in which antigen is delivered directly to GC B cells independent of the BCR, the authors showed that Tfh cells promote selection within the light zone. GC B cells capture antigen via the BCR and present the processed antigen on MHC complexes to Tfh cells. Higher BCR affinity is directly associated with greater antigen capture and leads to a higher density of peptide–MHC presentation on the surface of the B cell. This results in the greatest share of T cell help, which in turn drives selection.

However, we agree with the reviewer in that future models should consider in more detail how affinity translates into antigen uptake and presentation, including the effects of antibody feedback. We explicitly refer to this now in the discussion.

  1. Models 1-3 of B cell selection only differ in the way cell selected to leave the GC chooses between the plasma cell or memory cell fate. As such, the total number of output cells (MBC + PC) is the same in the 3 models, and the only measurable difference is the number of MBC versus PC in each model. This is reflected in Fig. 4E and F. Unfortunately, the existing data do not allow to confidently elect a winning model, with maybe the exclusion of 'Dynamic differentiation' (model 2), which seems to be disfavored by the measured output of MBCs.

  1. The delay of ~5 days corresponds to roughly 2-3 rounds of DZ-LZ B cell cycles. We believe that producing high-affinity antibodies takes several rounds, as it should take several rounds of division-selection for a clone to become dominant. Specifically, with an average of 3 divisions per DZ-LZ cycle, our model suggests that one round is not enough to achieve clonal dominance. We have added a comment in the manuscript to reflect this discussion.

Reviewer 3 Report

In the manuscript „Computational model reveals a stochastic mechanism behind germinal center clonal bursts” the authors present a quantitative stochastic model of the GC reaction to investigate the mechanism behind GC biology. The authors find that stochastic effects play an important role in the emergence of dominance.

The authors build on a previous multi scale GC model and add features to allow tracking of the effects of somatic hypermutation and BCR affinity over time.

The probabilistic model the authors show in Figure 1 is reasonable. The modified Gillespie algorithm allows to overcome the limitation of the amount of antigen captured. The authors use reasonable estimates for parameters based on existing literature to define GC initiation, centroblast division, centroblast migration, centrocyte antigen uptake and centrocyte-Tfh binding, centrocyte competition for Tfh help, centrocyte-Tfh spontaneous unbinding, centrocyte apoptosis, centrocyte recirculation and centrocyte exit.

I have 2 comments on these assumptions:

  1. The authors use 10BCR molecules which is published for mature B cells. GC B cells have lower BCR levels than mature B cells so I wonder whether this parameter needs to be adjusted.

  1. There is also evidence in the literature the GC reactions can occur in the absence of follicular dendritic cells (summarized in Shlomchik et al., 2019). I wonder whether the authors could include this in their model.

I think the parameters for cell fate and Tfh dynamics are very valid. I wonder whether the model would benefit from adding the effects of T follicular regulator cells?

Another thing to think about is that antigen becomes limiting over time (at least in the setting of non-replicating antigens like immunizations). Could the authors please comment on this.

In summary the current manuscript utilizes current literature to obtain reasonable estimates of values of all parameters used to feed the model. The model is in good agreement with experimental data. The model can help to focus experimental procedures as - for example – the model predicts that GC dynamics is most sensitive to Tfh mediated selection processes (found through stability analysis by the authors). The average affinity prediction over time is impressive as well as the GC output data which mimic experimental data after protein immunization. The model will help our better understanding in many areas of active experimental research like dynamics, fate, affinity maturation, clonal diversity and dominance. The model also allows for easy integration of new experimental data.

literature: Shlomchik, M. J., Luo, W. & Weisel, F. Linking signaling and selection in the germinal center. Immunol Rev288,49–63 (2019).

Author Response

  1. Mature B cells have been reported to carry up to 105 BCRs on their cell surface”. We are not aware of any paper that has estimated the number of BCRs on GC B cells surface, and we only use the number 105 to justify the assumption that a B cell can acquire and internalize many antigens at once. Indeed, although the number of BCRs in GC B cells is likely to be variable, our model does not explicitly use this count, and hence, our conclusions remain valid even if the number of receptors is demonstrated to be 2 orders of magnitude lower.

  1. Our model is not contingent on the presence of FDCs and could be used to simulate any other manner of acquiring antigen. The model is based on antigen acquisition from FDCs because this is widely accepted as the most common mechanism for antigen uptake. However, changing this part will not affect the main results, as the main factor that determines the outcome of the interaction with T cells is the amount of pMHC expressed.

  1. Our model can be extended in several ways, for instance, by including T follicular regulator cells and explicitly modelling their role in regulating excessive T follicular cells and GC B cell proliferation and promoting the selection of high-affinity B cells. These are very interesting aspects that we plan to explore in future work. We have added a paragraph in the discussion.

  1. Alternative quantitative work (Zhang & al) suggests that GC termination cannot be caused by limiting antigen alone, because the available antigen on FDCs decreases too slowly to be limiting. Thus, our model follows this assumption and assumes that there is sufficient antigen throughout the whole GC lifespan. As we mention in the discussion, antigen could be limiting if there is antibody feedback, but this hypothesis has not been experimentally verified yet.

Ref: Zhang & al, Germinal center B cells govern their own fate via antibody feedback, J. Exp. Medi, 2013.      

Reviewer 4 Report

The manuscript is a valuable contribution and well written and presented.

A GC model based on a Gillespie algorithm was implemented/extended.
The parameters of the model were optimized using public available
single cell sequencing and experimental kinetics data generated from GC.
The model was used to study assess qualitative and quantitative
the dynamics and kinetics of the components in GC (cell types and compartments), e.g., from the naive
B-cell entering the GC and the complex processes leading
to apoptosis, MBC or PC differentiation and clonal expansion.

The abstract appears weak and should contain more clearer the
explicit goals, results and conclusions of the manuscript.
Just one example "extensive quantitative characterizations" and
also describe the basic components of the model.

In the discussion some brief additional comments, comparisons
and information about previous work/literature
from other approaches and or similar models
would help to increase the value and novelty of the manuscript.

code availability planned?

line 316 The corresponding trees "are"

Table 1
section intracellular
NCDR & CDR "length"

Author Response

1) The abstract has been updated to address the reviewer’s comments.

2) The discussion section has been updated to address the reviewer’s comments (l. 618-620).

3)  The code is available at https://github.ibm.com/SysBio/Germinal-Center

Reviewer 5 Report

The manuscript titled "Computational model reveals a stochastic mechanism
behind germinal center clonal bursts" presented a computational model of the germinal centers of the human immune system using a stochastic methodology. The paper is very well written and of timeline and high interest, however there are revisions that need to be made before the paper is ready for publication.

1. Reactions. Lines 99-111 (short description) do not match with lines 134-177 (long description). Further, the short description seems to be wrong. Specifically lines 104, 106. This needs revision. I suggest that you fix the short description, but instead of bullet points, put the reactions in a table. You should also put the 'name' for each reaction in the table (a very short description of each reaction). Further, to aide in readers replicating your results, you should have a table of values for the reaction rates included with, or next to this table. Currently the values are found in table 1 is on line 383. It would be useful to move this table next to the reactions.

2. Constancy in naming of parameters. Please be sure the parameters and species are exactly the same in the short description, long description and figure 1. One example "r_FDC:CC" vs "r_FDC". The differing notation is confusing.

4. line 591 "The code is available freely on Github, along with all the data used to fit our model parameters" Please provide this url so that we may verify.

Author Response

1) and 2) The section lines 99-177 has been updated as requested by the reviewer

3)  The code is available at https://github.ibm.com/SysBio/Germinal-Center